# Ordered creation and motion of skyrmions with surface acoustic wave

Ruyi Chen[1], Chong Chen[1], Lei Han[1], Peisen Liu[1], Rongxuan Su[1], Wenxuan Zhu[1], Yongjian Zhou[1], Feng Pan[1] & Cheng Song ●[1] ✉

Magnetic skyrmions with a well-defined spin texture have shown unprecedented potential for various spintronic applications owning to their topologically non-trivial and quasiparticle properties. To put skyrmions into practical technology, efficient manipulation, especially the inhibition of skyrmion Hall effect (SkHE) has been intensively pursued. In spite of the recent progress made on reducing SkHE in several substituted systems, such as ferrimagnets and synthetic antiferromagnets, the organized creation and current driven motion of skyrmions with negligible SkHE in ferromagnets remain challenging. Here, by embedding the [Co/Pd] multilayer into a surface acoustic wave (SAW) delay line where the longitudinal leaky SAW is excited to provide both the strain and thermal effect, we experimentally realized the ordered generation of magnetic skyrmions. The resultant current-induced skyrmions movement with negligible SkHE was observed, which can be attributed to the energy redistribution of the system during the excitation of SAW. Our findings open up an unprecedentedly new perspective for manipulating topological solitons, which could possibly trigger the future discoveries in skyrmionics and spin acousto-electronics.

Efficient generation and manipulation of spin textures such as magnetic skyrmions has been a long-standing theme in the field of spintronics for their various potential applications, such as information storage[1–3], information processing[4,5] and neuromorphic computing devices[6,7]. In ferromagnetic systems, magnetic skyrmions were initially reported in chiral itinerant-electron magnet MnSi at low temperature[8]. Later, various methods, such as magnetic field[9–11], electric current/field[12–14], and thermal gradient[15–19] were proposed to generate skyrmions in magnetic multilayers with interfacial Dzyaloshinskii-Moriya interaction (DMI) at room temperature. Nevertheless, skyrmions generated in such studies are randomly distributed, making it challenging to use them as information carriers. The skyrmion Hall effect (SkHE) is another obstacle that restricts the essential transmission of skyrmions in devices, where skyrmions feel the Magnus force due to the finite topological charge[3,20,21]. Such Magnus force deflects the movement trajectory of magnetic skyrmions from the driving current, which is unfavorable for the storage stability and further device reliability.

Although several material systems, such as ferrimagnets[22–24] and synthetic antiferromagnets[25–27] have been demonstrated to possess skyrmions with reduced SkHE, the organized creation of skyrmions and current-driven motion with negligible SkHE in the prototypical ferromagnets still remain challenging, which has limited the applicability of skyrmions in practical devices.

A promising strategy for modifying magnetic textures at the nanoscale together with high controllability is to use strain. Surface acoustic waves are strain waves that can be excited through oscillating electric fields and propagate millimeter distances at the surface of piezoelectric materials[28–30]. By depositing magnetic films on the surface, the alternating strain generated in them can modify the magnetic interactions through the magnetoelastic effect, making it possible to manipulate the magnetic state[31]. Indeed, SAWs have already been used to induce magnetization oscillations[32,33], to assist the switching of the magnetic moments[34,35], and to control the dynamics of magnetic textures[31,36–42].

---

[1]Key Laboratory of Advanced Materials (MOE), School of Materials Science and Engineering, Beijing Innovation Center for Future Chip, Tsinghua University, Beijing 100084, China. ✉e-mail: songcheng@mail.tsinghua.edu.cn

Here, by fabricating an integrated SAW device, we report an experimental realization of the organization of magnetic skyrmions and the resultant current-induced skyrmions movement with negligible SkHE, simultaneously. We note that the comb-shaped interdigital transducers (IDTs) used in the measurements not only can generate thermal effect to assist the formation of magnetic skyrmions but can also excite the SAWs to induce alternating strain in magnetic films. In particular, the energy redistribution of the system caused by the strain gradient provides an effective tool to manipulate skyrmions. Hence, under the excitation of SAWs, skyrmions are initially created with a random distribution and then pushed towards the anti-nodes of the waves, exhibiting an ordered feature. Furthermore, theoretical simulations were performed to explain this pinning of skyrmions at the anti-nodes of SAWs based on the energy landscape. Our results complement the efficient manipulation of magnetic skyrmions in ferromagnets and may advance the further investigation of technologically relevant physics/device conceptions.

## Results

We start by discussing our approach to realize the organized motion of magnetic skyrmions in ferromagnets embedded into the SAW delay line. Two aspects of issues need to be prepared in our scenario. On one hand, the sample design, such as the magnetic multilayers with moderate perpendicular magnetic anisotropy (PMA), interfacial DMI, and film thickness play a significant role on the generation of magnetic skyrmions[43]. Here, the Co/Pd/Co/Pd/Co/Pt multilayer structure was chosen in our experiments not only for the

proper magnetic parameters and small dipolar fields in favor of skyrmion stability[11] but also for their potential to generate skyrmions by thermal effect[19]. On the other hand, to make use of the SAWs-induced magnetoelastic effect, we fabricated the delay line device which consists of two-port IDTs with a magnetic channel embedded in the cavity as shown in Fig. 1a. Here, a wire geometry was designed for the following dynamic measurements. To create magnetic skyrmions, a radio-frequency (RF) voltage was inputted into IDTs to excite the SAWs together with thermal effect. It is noted that since the strain gradient induced by SAWs is periodic with corresponding force vanishing at the anti-nodes of the wave[31], the generated skyrmions by thermal effect exhibit a more stable state at the anti-nodes of SAWs. Thus, by means of magneto-optical Kerr effect (MOKE) microscopy, we can directly observe the pinning of skyrmions in magnetic film with an ordered alignment. In addition, the transversal component during the current-driven motion of skyrmions should also be suppressed by SAWs, making it promising to eliminate the SkHE.

Stack structure of Co(0.3)/Pd(0.9)/Co(0.3)/Pd(0.9)/Co(0.3)/Pt(1.4) (units in nanometer) was deposited on 128°-rotated, Y-cut LiNbO$_3$ substrate via magnetron sputtering (Methods and Supplementary Fig. S1). In Fig. 1b, we present the temperature distribution image of the integrated device obtained by infrared camera (Supplementary Fig. S2). The width and gap of the fingers are both designed as 5 μm to excite a SAW with a well-defined wavelength, and the width of the magnetic channel is designed as 60 μm for the current-induced skyrmions movement measurements. By

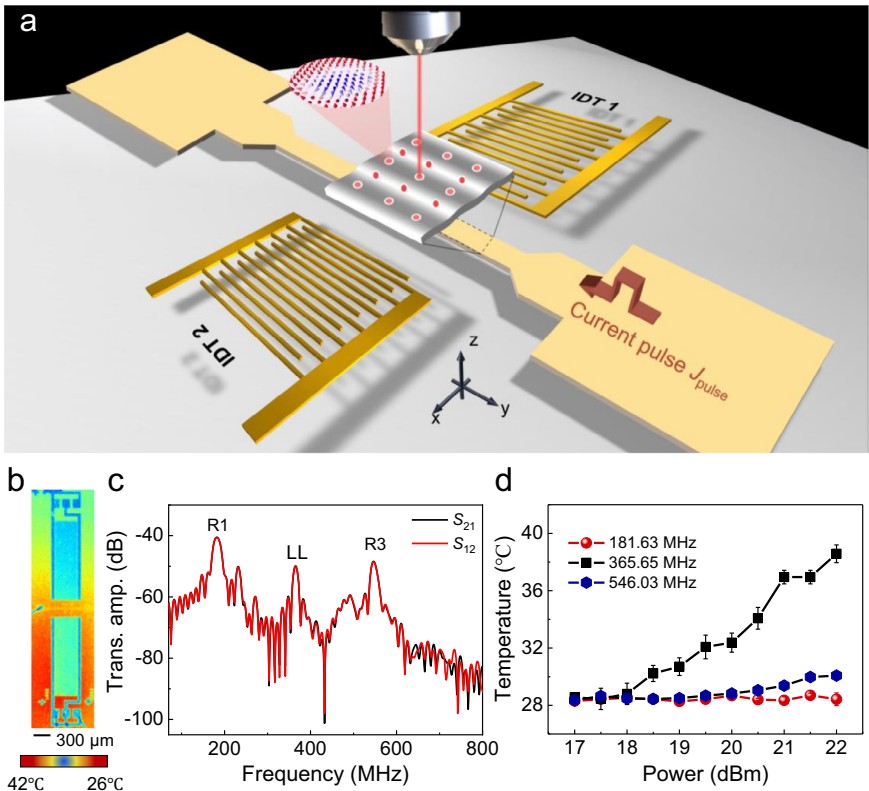

**Fig. 1 | Schematics of experimental setup and basic properties of surface acoustic wave devices. a** Schematic of the integrated device with a pair of IDTs on both sides of the magnetic multilayer channel. Néel-type skyrmions are generated at the antinode of SAW which can be observed via magneto-optical Kerr effect microscopy. Current pulses are applied to drive the movement of created skyrmions along the channel. **b** Temperature image of the integrated device obtained by the infrared camera when RF voltage of 365.65 MHz and 21 dBm are applied to

the IDT. **c** Typical SAW transmission spectra of the 5 μm wide delay line device with the magnetic channel placed between the two IDTs, which show three excited modes corresponding to the first (181.63 MHz), second (365.65 MHz) and third (546.03 MHz) harmonic. The spectra are obtained using RF signal of −5 dBm. **d** The average temperature at the center of magnetic channel under different frequencies as a function of applied power. The error bars correspond to the standard deviation.

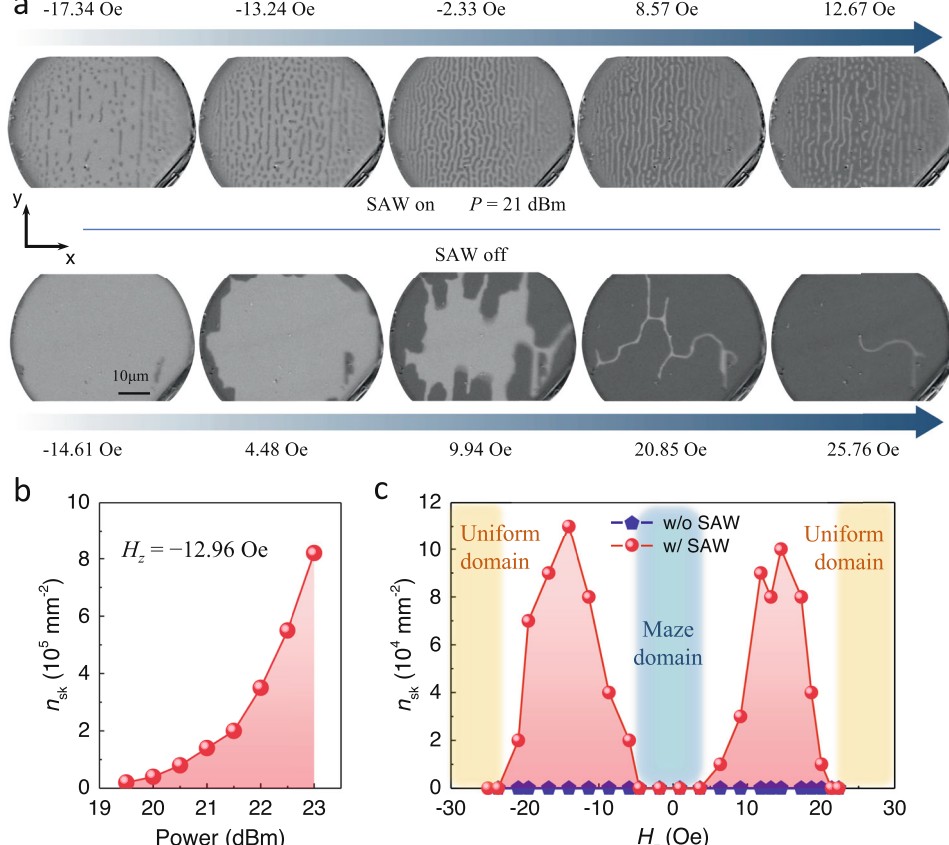

**Fig. 2 | Transformational dynamics and phase diagram of magnetic films with and without the excitation of SAWs. a** Consecutive MOKE images acquired from the Co/Pd/Co/Pd/Co/Pt multilayer under different perpendicular magnetic fields with (upper row) and without (bottom row) SAWs. Images taken at −17.34 Oe, −13.24 Oe, −2.33 Oe, 8.57 Oe and 12.67 Oe with the RF signal of $P$ = 21 dBm are shown in the upper column. Images taken at −14.61 Oe, 4.48 Oe, 9.94 Oe, 20.85 Oe and 25.76 Oe without SAW are shown in the bottom column. **b** The density of skyrmions ($n_{sk}$) as a function of the applied power $P$ under the magnetic field of −12.96 Oe. **c** The density of skyrmions phase diagram summarizing the evolution of different magnetic phase as a function of magnetic field $H_z$ with and without SAWs.

applying RF voltage with 365.65 MHz and 21 dBm to the IDT, the excitation of SAWs lead to the apparent increase of temperature which is favorable for the generation of skyrmions. Figure 1c shows the typical SAW transmission spectra of the delay line, which contain three excited modes corresponding to the first Rayleigh SAW (181.63 MHz, R1), the longitudinal leaky SAW (LLSAW) (365.65 MHz, LL)[44–46] and the third Rayleigh SAW (546.03 MHz, R3), confirmed by our finite element simulations (FEM) (Supplementary Fig. S3). The three peaks are determined by the sound velocity of the piezo-electric substrate and the geometry of IDTs. The similar features of the transmission spectrum are also observed in the FEM (Supplementary Fig. S4). We next estimate the thermal effect during the excitation of SAWs using the infrared camera. In Fig. 1d, we summarized the average temperature at the center of magnetic channel with respect to the three typical frequencies as a function of applied power. It is clear that the temperature in the magnetic films increases from 28.6 °C to about 38.6 °C as the increase of applied power from 17 dBm to 22 dBm at 365.65 MHz, where LLSAW goes deeply into the solid and radiates energy into the substrate[47]. Resultantly, there are large propagation losses in LLSAW, and the leaky power eventually dissipates in the form of heat, resulting in a remarkable temperature rise. In contrast, the temperatures in the magnetic film change much smaller (below 1 °C) at the same range of power at 181.63 MHz and 546.03 MHz. Therefore, we note that the LLSAW at 365.65 MHz provide both the strain and thermal effect in our delay line, paving the way for the later creation of organized skyrmions.

To validate our design, we next studied the evolution process of magnetic domains in the deposited multilayers with MOKE microscopy. We show in Fig. 2a the consecutive MOKE images acquired from the Co/Pd/Co/Pd/Co/Pt multilayer under different perpendicular magnetic fields with (upper row) and without (bottom row) SAWs. For the applied RF voltage of 365.65 MHz and 21 dBm along x-direction, intertwined maze domains were observed at the magnetic field of −2.33 Oe which evolve into the mixture of stripe domains and sky-rmions with increasing perpendicular magnetic field. The most strik-ing feature is that all of the generated skyrmions and maze domains align in the y-direction, exhibiting an ordered character. On the con-trary, for the frequency of 181.63 MHz and 546.03 MHz, no skyrmion or maze domain appears during varying the magnetic field. (Supple-mentary Fig. S5) As comparison, in the bottom row, only the coherent domain reversal was observed as we scanned the magnetic field from −14.61 Oe to 25.76 Oe without SAW. Thus, all of the results above indicate that (i) the generation of magnetic skyrmions is mainly induced by thermal effect, and (ii) the ordered alignments of sky-rmions are induced by the SAWs. Moreover, the phase diagram of magnetic films is further studied with and without SAWs. The density of skyrmions ($n_{sk}$) as a function of the applied power $P$ under the magnetic field of −12.96 Oe is shown in Fig. 2b. The skyrmions density increases gradually from 0 to $8.2 \times 10^5$ mm$^{-2}$ with increasing $P$ from 19 to 23 dBm. A skyrmions density phase diagram summarizing the evolution of coexisting stripe domain and skyrmions, maze domain, and uniform domain state as a function of the magnetic field with and without SAWs is shown in Fig. 2c. Three different magnetic phases can

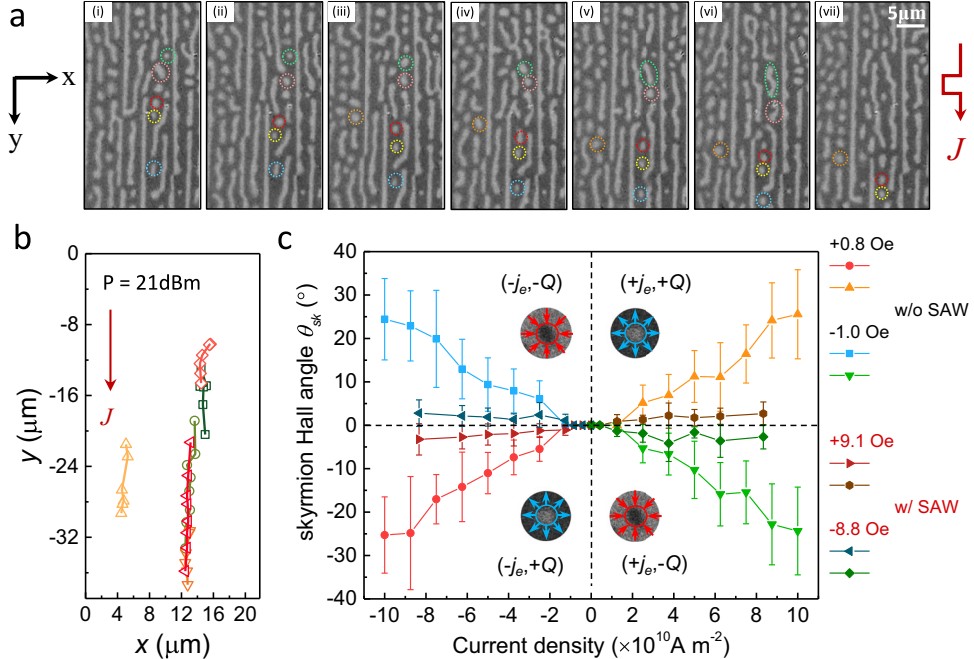

**Fig. 3 | Current-induced behavior of magnetic skyrmions and their skyrmion Hall effect with and without the excitation of SAW. a** Sequential MOKE images of magnetic domain patterns of Co/Pd/Co/Pd/Co/Pt multilayer under the current pulse applications at $H_z = 10.86$ Oe. Note that different skyrmions are characterized by different colors. Current density used to drive the skyrmion movement is $2.5 \times 10^{10}$ A m$^{-2}$ The duration of current pulses is 50 μs. **b** The movement trajectory of selected skyrmions in (**a**). **c** Phase diagram of the skyrmion Hall angle $\theta_{sk}$ as a function of current density measured with and without SAWs for skyrmions with topological charge $Q = +1$ and $Q = -1$. The skyrmions in the sample without SAW are generated through the thermal effect by fabricating a heater on the side of the channel. Each data point is an average of measurements obtained by tracking the movement of several independent skyrmions, with the error bars representing the standard deviation. The insets show the magnetic configurations of the skyrmions with opposite topological charge. Note that $H_z = +9.1$ Oe (−8.8 Oe) for $Q = +1$ (−1) with SAWs and $H_z = +0.8$ Oe (−1.0 Oe) for $Q = +1$ (−1) without SAW. The RF signal of $P = 21$ dBm was applied during all the current-driven measurements.

be distinguished under the excitation of SAWs scenario. When the magnetic fields are small ($|H_z| < 4$ Oe), only maze domain can be observed, which is due to the local instability of skyrmion phase under weak magnetic fields. When $4$ Oe $< |H_z| < 21$ Oe, the maze domain evolves into the mixture of skyrmions and stripe domains and the skyrmions density experience the first rises and then falls as increasing the magnetic fields. This can be attributed to the reason that the increasing Zeeman energy first overcomes the energy barrier to nucleate skyrmions and stripe domains and then destroies their stability as it is large enough. Finally, when the magnetic fields are large ($21$ Oe $< |H_z|$), the whole system transform into the uniform states and skyrmions disappear resulted from the forbidden of skyrmions by large Zeeman energy in the energy-stable ferromagnetic states[48]. In comparison, no skyrmion is observed in the film during scanning the magnetic field without SAW, indicating the necessary of SAW for the creation of magnetic skyrmions.

We now turn to the dynamic behavior of these orderly created skyrmions to see whether SAWs can suppress the transversal movement of current-induced skyrmions motion. Figure 3a displays a series of sequential MOKE images of magnetic domain patterns of Co/Pd/Co/Pd/Co/Pt multilayer under the current pulse of $2.5 \times 10^{10}$ A m$^{-2}$ at $H_z = 10.86$ Oe. The RF signal of $P = 21$ dBm was applied during the current-driven experiments. Here, different skyrmions are featured by different colors for the sake of distinction. Apparently, skyrmions move along the same direction of current pulses, confirming that these skyrmions are topologically protected with chiral Néel-type configuration[11] and their motion are governed by spin-orbit-torque originated from the heavy layer Pt[26]. We note that the elongation and disappearance of the bubbles marked by green and pink circles in (vi) and (vii) are most likely due to the pinning effect in the film which prevents the movement of skyrmions and eventually leads them to merge with other

domains. Strikingly, these skyrmions move almost in a straight line with negligible transverse component according to the movement trajectory presented in Fig. 3b, suggesting the reduced SkHE in our experiment. It is worth noting that not only skyrmions but also the stripe domains exhibit ordered alignment which can contribute to the inhibition of transversal movement of skyrmions[49,50]. As comparison, we prepared a sample with the same structure in which the mixed states of skyrmions and stripe domains generated through a heater (Supplementary Fig. S6). One can obviously see that the skyrmions and stripe domains are randomly distributed without any ordered states, suggesting the significant role of SAW on the ordered alignments of skyrmions and stripe domains. We also performed the micromagnetic simulations to study the current induced skyrmions motion with exciting SAWs. Consistent with the experimental results, skyrmions are first randomly distributed before exciting SAWs and then orderly aligned once the SAW is applied. Under the current pulses, skyrmions are moving in the current direction nearly in a straight line with negligible skyrmion Hall effect (Supplementary Fig. S7). Moreover, to quantitative demonstrate the influence of SAWs on SkHE, the current-driven skyrmion Hall angle $\theta_{sk} = \tan^{-1}(v_x/v_y)$ is evaluated, where $v_x$ and $v_y$ are velocity in the $x$ and $y$ directions[21]. In Fig. 3c, we summarize the phase diagram of the skyrmion Hall angle $\theta_{sk}$ as a function of current density with and without SAWs for skyrmions with topological charge $Q = +1$ and $Q = -1$. The skyrmions in the sample without SAW are generated through the thermal effect by fabricating a heater on the side of the channel (Supplementary Fig. S8). When the current density is small ($J < 1.25 \times 10^{10}$ A m$^{-2}$), skyrmions cannot be driven by the current for neither SAW nor the thermal induced cases. This is due to the fact that a certain large of current is needed to overcome the skyrmions pinning barrier. As increasing current density from $1.25 \times 10^{10}$ A m$^{-2}$ to $10 \times 10^{10}$ A m$^{-2}$,

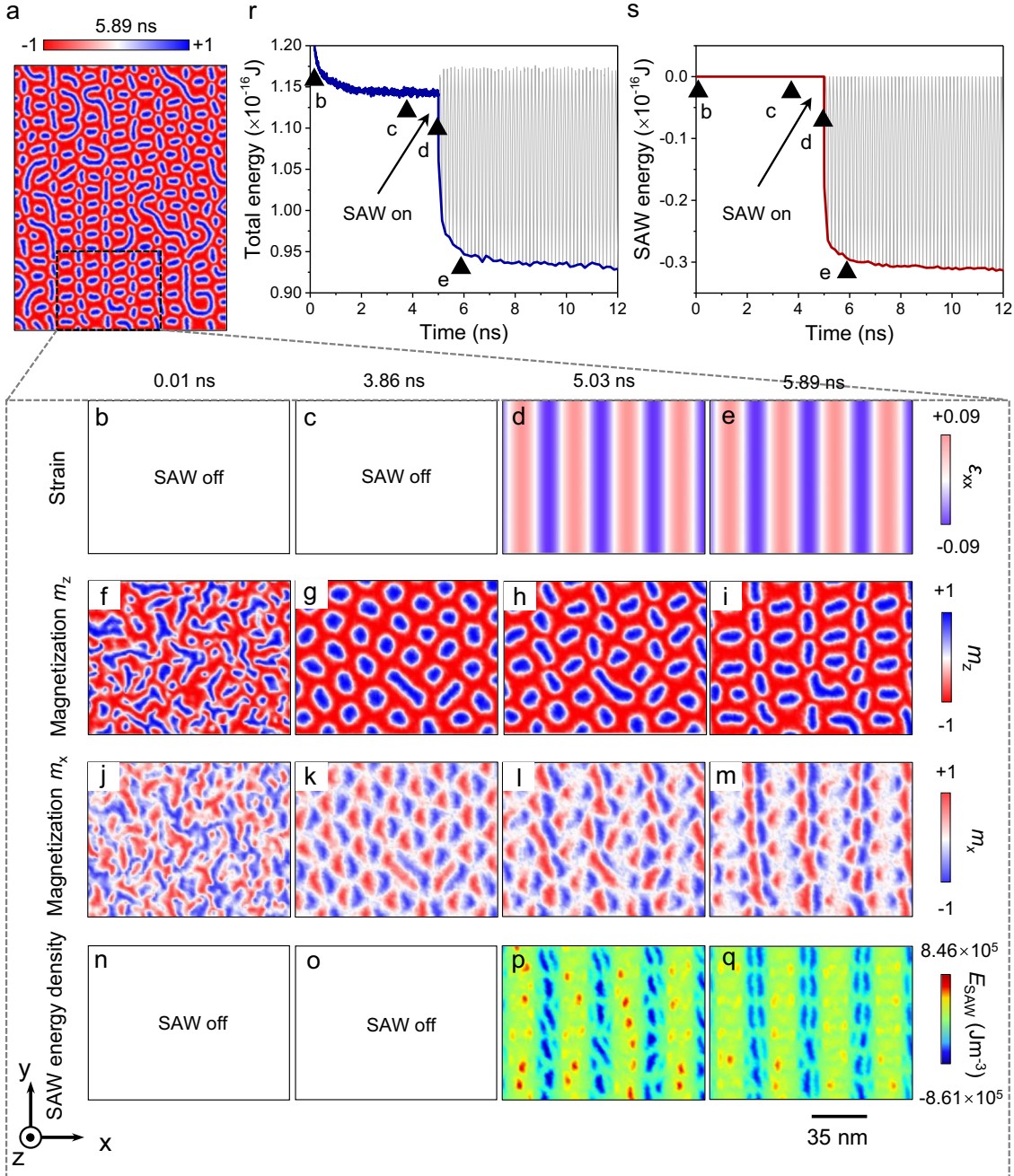

**Fig. 4 | Micromagnetic simulations of the SAW induced skyrmions alignment.** **a** The magnetization distribution of the simulated ferromagnetic layer at 5.89 ns. Here, the SAW was excited at 5 ns. The generated skyrmions and stripe domains are aligned in an organized way under the excitation of SAW. Color maps of the strain $\varepsilon_{xx}$ (**b–e**), magnetization $m_z$ (**f–i**), magnetization $m_x$ (**j–m**) and SAW energy density (**n–q**) for the selected area in (**a**) at different simulation time from 0.01 ns to 5.89 ns. The total energy (**r**) and SAW energy (**s**) vary with the simulation time. The solid triangles represent the time corresponding to the color maps of **b–e**.

skyrmions start to move and the corresponding skyrmion Hall angle increases rapidly from 0° to around 30° for the sample with thermal effect alone. In comparison, the $\theta_{sk}$ maintains at a small level (<7°) for the sample with SAWs. Such a distinct difference of these two scenarios suggests that SAWs play a significant role on the suppression of transversal movement during current-induced skyrmion motion which shows considerable potential to inhibit the SkHE in ferromagnets.

To discuss the microscopic origin of these experimental observations, we further carried out micromagnetic simulations in a quasi-two-dimensional system and studied the evolution process of spin textures. The magnetoelastic coupling energy can

be described as refs. 51–55:

$$E = b_1 \sum_i m_i^2 \varepsilon_{ii} + b_2 \sum_{i \neq j} m_i m_j \varepsilon_{ij} \tag{1}$$

where $b_1$ and $b_2$ are the magnetoelastic coefficients, and $\varepsilon_{ij}$ is the strain tensor induced by SAW. In the LLSAW, the only existing strain components are $\varepsilon_{xx}$, $\varepsilon_{xz}$ and $\varepsilon_{zz}$. Hence, the magnetoelastic coupling energy can be simplified as[28,53]:

$$E = b_1 \varepsilon_{xx} m_x^2 + b_1 \varepsilon_{zz} m_z^2 + 2b_2 \varepsilon_{xz} m_x m_z \tag{2}$$

In Fig. 4a, we present the magnetization $m_z$ distribution of ferromagnetic layer after the excitation of SAW where both skyrmions and stripe domains are aligned in an ordered way along the $y$-direction. Note that partial skyrmions and stripe domains are not perfectly aligned which may be ascribed to the existence of self-interactions between skyrmions and stripes. In fact, this phenomenon is also observed in our experiments as illustrated in Fig. 2a. For a deeper understanding, we then studied the entire evolution process of the selected area in Fig. 4a and the color maps of the strain, magnetization $m_z$, magnetization $m_x$ and SAW energy density at different simulation time ($t$) from 0.01 ns to 5.89 ns are shown in Fig. 4b–q. First, from 0.01 ns to 3.86 ns (without SAW), the film relaxes from a multi-domain state (Fig. 4f) to the skyrmion state (Fig. 4g) under the role of perpendicular magnetic anisotropy, interfacial DMI, external magnetic field, and the thermal fluctuations. Apparently, the generated skyrmions here are randomly distributed without any specific rules. Then, at $t$ = 5 ns, when the SAW was excited, spatially organized SAW energy density distribution appears (Fig. 4p and 4q). This organized SAW energy density is generated by the magnetoelastic coupling effect that varies with the alternating strain $\varepsilon_{xx}$. Therefore, the SAW related energy at skyrmion center can be written as $E_{skyrmion} = b_1 \varepsilon_{zz} m_z^2$, which exhibits a more stable state at the anti-nodes compared with the nodes of the wave. In this way, the generated skyrmions gradually move toward the antinode of the wave and finally constitute an organized alignment (Fig. 4i). To analyze the change in energy, we present in Fig. 4r and Fig. 4s the total energy and SAW energy vary as a function of simulation time, respectively. Before exciting SAW, the total energy decreases as the domains evolve from the initial state to the stable skyrmions state. Then, with the applied SAW, the total energy tends to reach a new state which varies with the oscillation of SAW energy (Fig. 4s). In other words, the emergent SAW redistributes the total energy of the system and makes skyrmions more stable at the anti-nodes of the wave, leading to their ordered alignment. Another evidence is that we also carried out the similar experiment in a delay line with the SAW frequency of 2.79 GHz while the created skyrmions are randomly distributed without ordered character. This may due to the wavelength of SAW is so small compared with the skyrmion size that there is no antinode available for skyrmions alignment (Supplementary Fig. S9).

In conclusion, we have demonstrated the ordered generation, manipulation and current-driven dynamics of magnetic skyrmions by the integrated SAW devices. We demonstrate that two aspects of issues need to be satisfied to realize the orderly alignment and movement of skyrmions. One is the magnetic structure design that we chose the Co/Pd/Co/Pd/Co/Pt multilayers with proper PMA, interfacial DMI, and film thickness. The other is the integrated delay line structure which can provide SAWs and thermal effect simultaneously. Thus, owing to the energy redistribution derived from the strain gradient during the excitation of SAWs, magnetic skyrmions are confined at the anti-nodes of the SAW and move nearly in a straight line under the current pulses. The unprecedented reduction of SkHE in ferromagnets is expect to drive the progress of skyrmion-based applications. Besides, our findings provide a completely new approach to manipulate topological solitons, which may advance the progress of further skyrmion-based spintronic devices.

## Methods
### Sample preparation
The films were deposited at room temperature onto 5 mm × 5 mm LiNbO₃ substrate for magnetic property measurements via d.c. magnetron sputtering with a base vacuum better than $8.0 \times 10^{-5}$ mTorr, and the working argon pressure was 3 mTorr. The complete stack structure used in the measurements is LiNbO₃ substrate/Co(0.3 nm)/Pd (0.9 nm)/Co(0.3 nm)/Pd(0.9 nm)/Co(0.3 nm)/Pt(1.4 nm).

### Device fabrication
To fabricate the delay line structure, we designed the IDTs with the width of 5 μm by using UV lithography followed by deposition of Ti (5 nm)/Al (100 nm) with electron beam evaporation. The distance of the two IDTs was 300 μm and each IDT had 70 pairs of single-type fingers. Here, the 128°-rotated, Y-cut LiNbO₃ substrate was selected for their piezoelectric property. Then, the magnetic films were patterned into the channel shape by standard optical lithography and lift-off process.

### RF and MOKE measurement
The substrate temperature was monitored by an infrared camera after the RF power was applied for 2 minutes. The transmission spectra between two IDTs were measured using a vector network analyzer and a RF signal generator was used to input the power during the experiments. The real-space skyrmions were observed using a polar MOKE microscope.

## Data availability
The data that support the findings of this study are available from the corresponding author upon request.

## Code availability
The codes used for the numerical calculations are available from the corresponding author upon reasonable request.

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

## Acknowledgements

This work was supported by the National Key R&D Program of China (Grant No. 2022YFA1402603), the National Natural Science Foundation of China (Grant Nos. 52225106 and 12241404), and the Natural Science Foundation of Beijing Municipality (Grant No. JQ20010). All of the funding receiver is the corresponding author, C.S.

## Author contributions

C.S. and R.C. designed the experiment. R.C., C.C., and Y.Z. grew the thin films and performed magnetization measurements. L.H. carried out the micromagnetic calculations. P.L. and R.S. performed the finite element simulations. F.P. gave suggestions on the experiments. C.S. supervised this study. All authors discussed the results and prepared the manuscript.

## Competing interests

The authors declare no competing interests.
