## [Peer Review File · Nature Communications]

Reviewers' Comments:

Reviewer #1:

Remarks to the Author:

In this manuscript, the authors have investigated the creation and motion of skyrmions by surface acoustic wave in the [Co/Pd] multilayers experimentally. The ordered magnetic skyrmions are generated in the multilayers under the effect of surface acoustic wave. It is also found that the current-induced skyrmions movement exhibits negligible skyrmion Hall effect. The work seems interesting. However, there are several experimental and theoretical studies on the creation and motion of skyrmions by surface acoustic wave. For example, the creation of skyrmion by SAW has been investigated by Yokouchi et al. [Nat. Nanotechnol. 15, 361–366 (2020)], and the motion of skyrmion driven by SAW has been investigated by Nepal et al. [Appl. Phys. Lett. 112, 112404 (2018)]. From this point of view, this manuscript is more like an extension of previous papers. It is not unexpected for the result although shown in a new system. I cannot recommend its publication in the Nature Communications in its present form. However, in case the manuscript is properly revised and more effort is paid to reveal the underlying mechanism and highlight the difference from previous papers, I would reconsider my decision. My detailed comments are as follows:

1. The authors show that the current-induced skyrmions movement exhibits negligible skyrmion Hall effect. And then they attribute this phenomenon to the energy redistribution of the system during the excitation of SAW. However, Fig. 3a shows the individual skyrmions are clamped by the strip domains along the current direction. The clamped skyrmions cannot move freely in the direction perpendicular to the current. Therefore, the negligible skyrmion Hall effect may have nothing to do with energy redistribution.
2. The author mentioned "the generation of magnetic skyrmions is the synergetic role of thermal effect and SAWs". As illustrated in Fig. S5, the skyrmion can be created by heater without SAW. Does this mean that the generation of skyrmions is related to heat, not SAW? I would like to see a more detailed explanation.
3. How is the temperature gradient in Fig. 1b generated? Is the temperature change related to the phase transition of ferromagnetic domains or only to the voltage? Why does the temperature rise fastest at 365.65 MHz?
4. There is a big difference in the magnitude of magnetic field with or without SAW in Fig. 3c. Why is that? Make sure that it is not a pinning effect caused by a strong magnetic field.
5. The author uses $\epsilon_{xx0}=0.09$ in Figure 4 and $\epsilon_{xx0}=0.125$ in the supplementary information, which are large strains. In fact, the strain should be less than 0.02.
6. The DMI constant seems to be larger than usual. Is there any reference material to support it?
7. Due to the Poisson effect, ϵ_{yy} will also be generated when there is ϵ_{xx} in the material, so the energy term of ϵ_{yy} cannot be ignored during the simulation. This simplification in the paper can indeed reduce the amount of calculation, but it will actually cause the results to deviate from the real situation.

Reviewer #2:

Remarks to the Author:

The paper by Chen et al. presents an organized creation and current-driven motion of skyrmions with negligible skyrmion Hall effect (SkHE) through the application of surface acoustic waves (SAWs). Micromagnetic simulations are performed to show the ordered alignment of skyrmions and stripe domains originating from the energy redistribution through SAWs. I think this work will pave a new way to SAW based skyrmionic devices, and it is an important contribution to the field of spintronics. However, before it is suitable for publication several important issues need to be addressed.

Important Issues:

- 1) The authors need to clarify which kind of SAWs are used in the experiments, propagating or standing SAWs. It is confusing since from Fig. 1, the propagating SAWs are used. While, from the description of "anti-node" and the expression of strain matrix used in the simulations, a standing SAWs form is used. If it is standing SAWs, a description about its generation, and also the temperature distribution should be added in details

2) In Fig. 3a, an ordered mixture of skyrmions and stripe domains is shown with the excitation of SAWs, and a negligible SkHE is observed compared to the skyrmions in Fig. S5. The skyrmions in Fig. 3a are next to the stripe domains, the authors need to address its influence on the SkHE, because a previous paper claimed that stripe domains can guide the skyrmion motion (Phys. Rev. Applied 18, 024030 (2022)). In addition, thermal gradient generated by propagating SAWs can also drive skyrmions and may influence its transverse motion. In short, the reduction of SkHE from the nearby stripe domains and thermal gradient need to be clarified.

Other Issues

- 1) In this work, the second harmonic waves (365.65MHz) are utilized to control skyrmions. According to theories, only odd harmonics can exist for the configuration of IDTs used here (D. Morgan, surface acoustic wave filters, Elsevier Ltd (2007)). This should be clarified.
- 2) "The skyrmions density increases gradually from 0 to $10 \times 10^5 \text{ mm}^{-2}$ with increasing P...", the density is high considering the skyrmion size in micrometer.
- 3) In page 8 "evolves into the mixture of skyrmions and stipe domains...", "stipe" should be "stripe".
- 4) In this work, obvious thermal effects are found and utilized to assist the skyrmion creation. However, skyrmion creation through SAWs performed by T. Yokouchi et al. (Nat. Nanotechnol. 15, 361–366 (2020)) shows that very low temperature change is observed when RF voltages are inputted into IDTs. What's possible reason for this difference?
- 5) The pulse width of driving current and the velocity of skyrmion motion should be added in the manuscript.

Reviewer #3:

Remarks to the Author:

In this work, the authors demonstrate the ordered formation of skyrmions and the suppression of the skyrmion Hall angle via the application of surface acoustic waves (SAWs). The reduction of the skyrmion Hall angle is one of the main topics in the skyrmion research. Recently, there are several ways to reduce the skyrmion Hall angle have been proposed and experimentally demonstrated. Since utilizing SAWs to achieve this is a novel way, this work has a great impact on spintronics. However, as described below, there are several concerns and uncertainty in the present form of the manuscript, some of which may affect the main conclusion. Hence, if the authors can clarify the following concerns, I can recommend the publication.

1. Did the authors use propagating SAWs or standing SAWs? Judging from Fig. 1(a) and (b), it seems propagating SAWs were used. If this is the case, what determines the positions of anti-nodes (i.e. the position of the skyrmion array?). In general, the positions of the anti-nodes of propagating SAWs move with time.

2. Figure 3 shows the skyrmion Hall angle with and without applying SAWs. However, the initial conditions for these experiments differ. When both SAWs and the current are applied, skyrmions and stripe domains coexist in the initial state (Fig. 3a). In contrast, when only the current is applied, only skyrmions exist in the initial state (Fig. S5). It is important to note that stripe domains can affect the skyrmion motion and reduce the skyrmion Hall angle [see PRB 101, 2144362 (2020).].

Hence, for a fairer comparison, the same initial state (hopefully initial state accommodating only skyrmions) should be used. For example, do the skyrmions created by the heater show the small skyrmion Hall angle if the current and SAW are applied simultaneously? If SAWs reduce the skyrmion angle as the authors concluded, the skyrmions created by the heater must show the small skyrmion Hall angle when both SAWs and current are applied.

3. In Fig. 3a, the authors show the snapshots of the magnetic texture when the current is applied. As can be seen from Fig. 3a, many bubbles (represented by red, orange, yellow, and blue circles) move along the current, supporting the authors' assertion that these bubbles are topologically

nontrivial (i.e. skyrmion). However, a few bubbles (green and pink circles) are elongated (vi) and disappear (vii), which indicates these bubbles are topologically trivial (i.e. not skyrmion), because the topologically trivial bubbles are elongated or shrink under the current [Science 349, 283 (2015)]. Hence, a few trivial bubbles are probably created by SAWs. While this point does not affect the authors' main conclusion, the authors should discuss this point more carefully in the main text.

4.The authors have simulated the current driven motion of skyrmions with SAW, which seems qualitatively consistent with the experiment results (Fig. S7). However, I cannot find any explanation or citation of this result in the main text. If the result shown in Fig. S7 has not been described in the main text, it should be mentioned.

5.How much is the gap size of the IDT used for exciting SAW with 2.79 GHz (Fig. S6)?

Response Letter of NCOMMS-22-52692-T

We very much appreciate the positive evaluations of our manuscript (NCOMMS-22-52692-T) by Reviewer #1: “The work seems interesting.”, Reviewer #2: “I think this work will pave a new way to SAW based skyrmionic devices, and it is an important contribution to the field of spintronics” and Reviewer #3: “Since utilizing SAWs to achieve this is a novel way, this work has a great impact on spintronics.”. They clearly spent much time reading our manuscript and we address the issues raised by them point by point below.

Amendments of our revised manuscript are summarized below in bold face style. The main modifications include:

1. We gave more in-depth explanations of the excitation of SAWs in our experiments, added some related simulation results and discussed the origin of thermal effect in the frequency of 365.65 MHz in both the main text and supplementary information. The related references are also added.

2. We added the discussions and contrast experimental results to confirm that SAWs are the most fundamental reason that leads to the ordered alignment of skyrmions and stripe domains, eventually inhibiting the transversal motion of skyrmions.

3. We revised the description of the generation of skyrmions in our experiments.

4. We counted a much wider area of the film and recalculate the density of skyrmions.

5. We discussed the reason that lead to the elongation and disappearance of the bubbles during the current induced motion in the main text.

6. We added the discussion about the simulated results of the current driven motion of skyrmions with SAW in the main text.

7. We added the velocity of skyrmions motion in the supplementary figure S10.

8. We added the width and gap of the fingers used to excite SAW with 2.79 GHz in the supplementary information.

9. We added some references to support the parameters used in our micromagnetic simulations in supplementary note 1.

10. We modified the typos and added the pulse duration of current in the main text.

Response to Reviewer # 1

In this manuscript, the authors have investigated the creation and motion of skyrmions by surface acoustic wave in the [Co/Pd] multilayers experimentally. The ordered magnetic skyrmions are generated in the multilayers under the effect of surface acoustic wave. It is also found that the current-induced skyrmions movement exhibits negligible skyrmion Hall effect. The work seems interesting. However, there are several experimental and theoretical studies on the creation and motion of skyrmions by surface acoustic wave. For example, the creation of skyrmion by SAW has been investigated by Yokouchi et al. [Nat. Nanotechnol. 15, 361–366 (2020)], and the motion of skyrmion driven by SAW has been investigated by Nepal et al. [Appl. Phys. Lett. 112, 112404 (2018)]. From this point of view, this manuscript is more like an extension of previous papers. It is not unexpected for the result although shown in a new system. I cannot recommend its publication in the Nature Communications in its present form. However, in case the manuscript is properly revised and more effort is paid to reveal the underlying mechanism and highlight the difference from previous papers, I would reconsider my decision.

Response: We are very grateful to the referee's evaluation "The work seems interesting.". As the referee mentioned, the interactions between skyrmions and SAW have been studied in previous works, which include the experimentally generation of skyrmions and theoretically predicted motion of skyrmions by SAW. However, there are several essential issues remain to be demonstrated: 1) Whether the skyrmions generated by SAW can be driven by current in experiment? 2) Whether the transversal movement of skyrmions (skyrmion Hall effect) can be inhibited by SAW? 3) How to design a suitable sample structure to investigate the effect of SAW on skyrmions? Our manuscript fills in these gaps and these are exactly the differences between our work and previous papers. In fact, considering the creation and stabilization of skyrmions have been realized in different systems, to make use of magnetic skyrmions in practical devices, the next concern is how to control and manipulate the skyrmions effectively. Our findings of ordered creation and motion of skyrmions with SAW here represent an important step towards manipulating skyrmions in ferromagnet and bringing skyrmions closer to practical applications. Besides, according to the referee's suggestions and requirements, we further revised our manuscript and gave more in-depth discussions in the revised version.

Q1: The authors show that the current-induced skyrmions movement exhibits negligible skyrmion Hall effect. And then they attribute this phenomenon to the

energy redistribution of the system during the excitation of SAW. However, Fig.3a shows the individual skyrmions are clamped by the stripe domains along the current direction. The clamped skyrmions cannot move freely in the direction perpendicular to the current. Therefore, the negligible skyrmion Hall effect may have nothing to do with energy redistribution.

Answer: We agree with the referee that the stripe domains contribute to the inhibition of transversal motion of skyrmions. In our work, it is noted that both skyrmions and stripe domains are orderly aligned under the excitation of SAW as shown in figure 3a. This means not only skyrmions but also the stripe domains show more stable state at the anti-nodes of waves as discussed in the micromagnetic simulation part. Thus, it can be seen that the energy redistribution caused by SAW is the most fundamental reason that leads to the ordered alignment of skyrmions and stripe domains and eventually causes the negligible skyrmion Hall effect. To further verify the importance of SAWs to the ordered alignment feature, we showed the mixed states of skyrmions and stripe domains generated by the heater in figure S6 as comparison. One can obviously see that the skyrmions and stripe domains are randomly distributed without any ordered states, suggesting the significant role of SAW on the ordered alignments of skyrmions and stripe domains. In the main text, we added the discussions in page #10 line #6 from the bottom: **It is worth noting that not only skyrmions but also the stripe domains exhibit ordered alignment which can contribute to the inhibition of transversal movement of skyrmions^{41,42}. As comparison, we prepared a sample with the same structure in which the mixed states of skyrmions and stripe domains generated through a heater (Supplementary Fig. S6). One can obviously see that the skyrmions and stripe domains are randomly distributed without any ordered states, suggesting the significant role of SAW on the ordered alignments of skyrmions and stripe domains.** The added figure S6 is shown as follows:

Fig. S6 | MOKE image for the mixed states of skyrmions and stripe domains generated by thermal effect in the sample of Co/Pd/Co/Pd/Co/Pt. Scale bar, 10 μm .

Q2: The author mentioned “the generation of magnetic skyrmions is the synergetic role of thermal effect and SAWs”. As illustrated in Fig. S5, the skyrmion can be created by heater without SAW. Does this mean that the generation of skyrmions is related to heat, not SAW? I would like to see a more detailed explanation.

Answer: In fact, both thermal effect (Nat. Electron. 3, 672–679 (2020), Adv. Funct. Mater. 32, 2111906 (2022)) and SAW (Nat. Nanotechnol. 15, 361–366 (2020)) can be used to create skyrmions. In our experiment, both SAW (figure 1c) and thermal effect (figure 1d) were detected at the applied frequency of 365.65 MHz. Specifically, considering the fact that no skyrmion or maze domain appears during varying the magnetic field for the frequency of 181.63 MHz and 546.03 MHz (Supplementary Fig. S5), we conclude that the thermal effect plays a more important role on the generation of skyrmions and SAWs are mainly used to make skyrmions orderly aligned. To make the statement more accuracy, we modified the sentence in page #7 line #6 from the bottom: **Thus, all of the results above indicate that (i) the generation of magnetic skyrmions is mainly induced by thermal effect, and (ii) the ordered alignments of skyrmions are induced by the SAWs.**

Q3: How is the temperature gradient in Fig. 1b generated? Is the temperature change related to the phase transition of ferromagnetic domains or only to the voltage? Why does the temperature rise fastest at 365.65 MHz?

Answer: The reviewer raised a very good point and remind us to elucidate the

temperature increasing in our SAW delay line more clearly. When an RF voltage (365.65 MHz) is applied to the IDT, the temperature at the surface of the substrate increases induced by SAW and a longitudinal temperature gradient ∇T_x emerges naturally due to the unilateral applying. As mentioned by the reviewer, we have carefully checked the apparent temperature rising at 365.65 MHz, where the longitudinal leaky SAW (LLSAW) has been excited confirmed by our FEM simulations (Fig. S3). In contrast to the Rayleigh SAW (181.63 and 546.03 MHz), whose strain and energy concentrate well at the surface, LLSAW goes deeply into the solid and radiates energy into the substrate (Appl. Phys. Lett. 86, 024101 (2005), IEEE Ultrasonics Symposium Proceedings 1, 287-292 (1994), IEEE Ultrasonics Symposium Proceedings 1, 157-160 (1997)). Resultantly, there are large propagation losses in LLSAW, and the leaky power eventually dissipates in the form of heat, resulting in a remarkable temperature rise (IEEE Transactions on Ultrasonics, Ferroelectrics, and Frequency Control 56, 2686-2692 (2009)). More quantitatively, the dissipated power P_{diss} can be calculated by means of S parameters:

$$P_{\text{diss}} = P_{\text{in}}(1 - |S_{21}|^2 - |S_{11}|^2)$$

where P_{in} is the input power by the RF signal generator. LLSAW and Rayleigh SAW have comparable S_{21} , but the S_{11} of LLSAW is significantly smaller, resulting in larger P_{diss} and more apparent thermal effect. Therefore, this large temperature increasing is mainly ascribed to the energy dissipation of LLSAW, which is excited by the RF voltage. We have mentioned this point and revised the corresponding statements in the main text in Page #5 Line #2 **“Figure 1c shows the typical SAW reflection spectrum of the delay line, which contains three excited modes corresponding to the first Rayleigh SAW (181.63 MHz, R1), the longitudinal leaky SAW (LLSAW) (365.65 MHz, LL) and the third Rayleigh SAW (546.03 MHz, R3), confirmed by our finite element simulations (FEM) simulations (Supplementary Fig. S3)”** and Page #5 Line #9 from the bottom **“It is clear that the temperature in the magnetic films increases from 28.6 °C to about 38.6 °C as the increase of applied power from 17 dBm to 22 dBm at 365.65 MHz, where LLSAW goes deeply into the solid and radiates energy into the substrate³⁹. Resultantly, there are large propagation losses in LLSAW, and the leaky power eventually dissipates in the form of heat, resulting in a remarkable temperature rise. In contrast, the temperatures in the magnetic film change much smaller (below 1 °C) at the same range of power at 181.63 MHz and 546.03 MHz. Therefore, we note that the LLSAW at 365.65 MHz provide both the strain and thermal effect in our delay line, paving the**

way for the later creation of organized skyrmions.”. The added figure S3 is shown as follows:

Fig. S3 | Finite element simulations of the SAW modes based on 128° Y- 90° X LiNbO₃ substrate with $\lambda = 20 \mu\text{m}$. **a**, The simulated device admittance curve. There are three resonance peaks with frequencies around 179, 319 and 527 MHz, respectively. The third peak is relatively weak (see the inset). **b**, The deformation shape for the three SAW modes at resonant frequency. From left to right, they are the first Rayleigh SAW (R1), the longitudinal leaky SAW (LL) and the third Rayleigh SAW (R3). **c,d**, The distribution of three normalized particle displacement components along the white dashed line for R1 (c) and LL (d). The insets show the local magnification of the deformation and white dotted lines extending from the electrode surface to the inside of the substrate. The distance from the electrode surface d is normalized through dividing by the wavelength λ . The propagation direction of SAW is defined as the $+x$ -axis. The displacements of R1 concentrate well at the surface ($1 \sim 2$ times of λ), while those of LL go deeply into the substrate.

Q4: There is a big difference in the magnitude of magnetic field with or without SAW in Fig. 3c. Why is that? Make sure that it is not a pinning effect caused by a strong magnetic field.

Answer: The difference in the magnitude of magnetic field to generate skyrmions for the two cases can be understood that the methods used to create skyrmions are totally different. For the case without SAW, the skyrmions are generated via thermal effect by fabricating a heater on the side of the channel. Combining the direct current applied through the heater and the perpendicular magnetic field, skyrmions can be stabilized. On the other hand, skyrmions are created by means of the excitation of SAW for the case with SAW. Besides the thermal effect caused by SAW, the SAW itself also contribute to the generation of skyrmions (Nat. Nanotechnol. 15, 361–366 (2020)). Therefore, the magnetic fields needed to stabilize skyrmions are different in these two cases.

Q5: The author uses $\epsilon_{xx}=0.09$ in Figure 4 and $\epsilon_{xx0}=0.125$ in the supplementary information, which are large strains. In fact, the strain should be less than 0.02.

Answer: We agree with the referee that the strain used in the simulations is relatively large for the sake of demonstrating the skyrmions alignment observed in the experiments. In fact, we also performed the micromagnetic simulations for a smaller strain ($\epsilon_{xx0} = 0.04$), in which the SAW-driven ordered alignment of skyrmions can also be observed. Analogously, the amplitude (strain) of 0.06 in the SAW has been adopted in their simulations to explain the ordered domain patterns in a previous work. (Appl. Phys. Lett. 120, 252402 (2022)). The simulated results for the $\epsilon_{xx0} = 0.04$ in our system are shown as follows:

Fig. R1 | Simulated colour maps of the strain ϵ_{xx} (top) and magnetization m_z (bottom) for $\epsilon_{xx0} = 0.04$.

Q6: The DMi constant seems to be larger than usual. Is there any reference material

to support it?

Answer: The DMI constant was set to be 4 mJ m^{-2} , which is consistent with many former works to simulate the Co/Pt based multilayers. (Nat. Nanotech. **8**, 839 (2013): $0 - 9 \text{ mJ m}^{-2}$, Nat. Commun. **7**, 10293 (2016): $0 - 6 \text{ mJ m}^{-2}$, Nat. Mater. **16**, 898 (2017): $2 - 6 \text{ mJ m}^{-2}$). We have added these references in our simulation method. In fact, the setting of DMI constant does not have essential influence on the simulations of SAW-driven ordered alignment of skyrmions. For example, SAW-driven ordered alignment of skyrmions can also be achieved under DMI of 3.0 mJ m^{-2} . The corresponding simulation result is shown as follows:

Fig. R2 | Simulated colour maps of the magnetization m_z for the $D = 3.0 \text{ mJ m}^{-2}$.

Q7: Due to the Poisson effect, e_{yy} will also be generated when there is e_{xx} in the material, so the energy term of e_{yy} cannot be ignored during the simulation. This simplification in the paper can indeed reduce the amount of calculation, but it will actually cause the results to deviate from the real situation.

Answer: We thank the reviewer for this constructive comment and acknowledge that there is a Poisson effect in solids, as the reviewers pointed out. But for surface waves in solids, the situation seems more complicated. In LLSAW, the dominating longitudinal displacement component (x direction) decays exponentially into the depth of the substrate, while the transverse component (y direction), as well as ϵ_{yy} , is zero (Appl. Phys. Lett. **82**, 3351 (2003), Appl. Phys. Lett. **86**, 024101 (2005)). We have also added the FEM simulations for LLSAW propagating along rotated $90^\circ X$ on $128^\circ Y$ -cut LiNbO_3 substrate in Fig.S3. In Fig. S3d, we define the propagation direction as the x -axis and the out-of-plane direction as the z -axis. The particle has a large displacement u_x but zero u_y . Meanwhile, u_z although also relatively small, is not zero and it changes periodically along z -axis, indicating that this acoustic wave leaks into the substrate. Therefore, the only existing strain components are ϵ_{xx} , ϵ_{zz} and ϵ_{xz} in our LLSAW, and ϵ_{yy} is insignificant for magnetoelastic coupling energy in our micromagnetic simulations.

Response to Reviewer # 2

The paper by Chen et al. presents an organized creation and current-driven motion of skyrmions with negligible skyrmion Hall effect (SkHE) through the application of surface acoustic waves (SAWs). Micromagnetic simulations are performed to show the ordered alignment of skyrmions and stripe domains originating from the energy redistribution through SAWs. I think this work will pave a new way to SAW based skyrmionic devices, and it is an important contribution to the field of spintronics. However, before it is suitable for publication several important issues need to be addressed.

Response: We are grateful to the reviewer's support and the positive evaluation of our work.

Q1: The authors need to clarify which kind of SAWs are used in the experiments, propagating or standing SAWs. It is confusing since from Fig. 1, the propagating SAWs are used. While, from the description of "anti-node" and the expression of strain matrix used in the simulations, a standing SAWs form is used. If it is standing SAWs, a description about its generation, and also the temperature distribution should be added in details.

Answer: The SAWs used in our experiment are formed through the coherent superposition of longitudinal leaky SAWs (LLSAWs), which exhibit the feature of standing wave. The excitation of LLSAWs has been discussed in detail in the response to Q3. According to our FEM simulations (Fig. S4), the process of coherent superposition can be described as follows: The LLSAW is excited when an RF voltage is applied to IDT 1 (Fig. S4a). Then, LLSAW propagates along the +x-axis and coherently superimposed with the reflected wave from the IDT 2 to form a bulk longitudinal wave (leakage along the -z-axis). Meanwhile, the strain distribution on the substrate surface has the characteristics of standing waves, i.e. there are some locations where the strain is always zero (nodes) or largest (anti-nodes), as shown in Fig. S4d. Along with these anti-nodes, the position of the skyrmion array is determined. The remarkable temperature increasing is attributed to the energy dissipation of LLSAW, we will discuss this in detail in our response to Q6. The standing waves on the surface have no significant thermal effect and the temperature distribution is derived from leaky waves as shown in Fig. 1b. We mentioned this point in Supplementary Note 2 as **"The SAW admittance and transmission spectrums are simulated by commercial COMSOL 6.0 Multiphysics platform using piezoelectricity component based on solid**

mechanics and electrostatics. To reduce computational load, we build 2.5D resonator and acoustic delay line (ADL) model and set a proper thickness in plane. As for resonator simulation, the model possesses following geometry: wavelength $\lambda=20\ \mu\text{m}$, the lithium niobate (LN) substrate was set to be 4.5λ and 1λ perfect match layer was added to the bottom for capturing losses related to bulk wave radiation. In order to simplify the calculation time, only a period of IDT is considered, and periodic boundary condition has been set in both the x- and y-directions to simulate the infinite length. As for ADL simulations, the distance of the two IDTs is $200\ \mu\text{m}$, and each IDT has 20 pairs of single-type fingers. In each IDT, the width and gap of the fingers are both set to $5\ \mu\text{m}$, defining the fundamental wavelength of the SAW to be $20\ \mu\text{m}$. Perfectly matched layers (PML) are placed around the LN cells for the same effect as resonator. The frequency domain studies are carried out to calculate the Y- and S-parameters of resonator and ADL, respectively. The calculated S_{21} of this device showing three transmission maxima, in coincidence with those of the device used in the experiment. For the second peak which is used in our experiments, the LLSAW is excited when an RF voltage is applied to IDT 1 (Fig. S4a). Then, LLSAW propagates along the +x-axis and coherently superimposed with the reflected wave from the IDT 2 to form a bulk longitudinal wave (leakage along the -z-axis). Meanwhile, the strain distribution on the substrate surface has the characteristics of standing waves, i.e. there are some locations where the strain is always zero (nodes) and where the strain is largest (anti-nodes), as shown in Fig. S4d. Along with these anti-nodes, the position of the skyrmion array is determined.”. In addition, we modified the schematic in figure 1a and the revised figure S4 is shown as follows:

Fig. S4 | Finite element simulations of the SAW transmission spectrum in a delay line based on 128° Y- 90° X LiNbO₃ substrate. **a,b**, Device geometry and corresponding coordinate system. The distance of the two IDTs is 300 μm , and each IDT has 20 pairs of single-type fingers. In each IDT, the width and gap of the fingers are both set to 5 μm , defining the fundamental wavelength of the SAW to be 20 μm . **c**, The calculated S_{21} of this device, showing three transmission maxima, in coincidence with those of the device used in the experiment. **d**, The strain distribution near the surface of LiNbO₃ substrate in the xOz plane, showing the characteristics of standing waves.

Q2: In Fig. 3a, an ordered mixture of skyrmions and stripe domains is shown with the excitation of SAWs, and a negligible SkHE is observed compared to the skyrmions in Fig. S5. The skyrmions in Fig. 3a are next to the stripe domains, the authors need to address its influence on the SkHE, because a previous paper claimed that stripe domains can guide the skyrmion motion (Phys. Rev. Applied 18, 024030 (2022)). In addition, thermal gradient generated by propagating SAWs can also drive skyrmions and may influence its transverse motion. In short, the reduction of SkHE from the nearby stripe domains and thermal gradient need to be clarified.

Answer: We agree with the referee that the stripe domains contribute to the inhibition of transversal motion of skyrmions and this has been discussed in previous work (Phys. Rev. B 101, 214432 (2020), Phys. Rev. Applied 18, 024030 (2022)). In our paper, it is noted that both skyrmions and stripe domains are orderly aligned under the excitation of SAW as shown in figure 3a. This means not only skyrmions but also the stripe domains show more stable state at the anti-nodes of waves as discussed in the micromagnetic simulation part. Thus, it can be seen that the energy redistribution caused by SAW is the most fundamental reason that leads to the ordered alignment of skyrmions and stripe domains and eventually causes the negligible skyrmion Hall effect. On the other hand, the temperature needed to form skyrmions is not high so that the thermal effects generated by both SAW and heater are small. In this case, the influence of transversal motion by thermal gradient can be ignored which can be verified in figure 2a and figure S8.

To further verify the importance of SAWs to the ordered alignment feature, we showed the mixed states of skyrmions and stripe domains generated by the heater in figure S6 as comparison. One can obviously see that the skyrmions and stripe domains are randomly distributed without any ordered states, suggesting the significant role of SAW on the ordered alignments of skyrmions

and stripe domains. In the main text, we added the discussions in page #10 line #6 from the bottom: **It is worth noting that not only skyrmions but also the stripe domains exhibit ordered alignment which can contribute to the inhibition of transversal movement of skyrmions^{41,42}. As comparison, we prepared a sample with the same structure in which the mixed states of skyrmions and stripe domains generated through a heater (Supplementary Fig. S6). One can obviously see that the skyrmions and stripe domains are randomly distributed without any ordered states, suggesting the significant role of SAW on the ordered alignments of skyrmions and stripe domains.** The added figure S6 is shown as follows:

Fig. S6 | MOKE image for the mixed states of skyrmions and stripe domains generated by thermal effect in the sample of Co/Pd/Co/Pd/Co/Pt. Scale bar, 10 μm .

Q3: In this work, the second harmonic waves (365.65MHz) are utilized to control skyrmions. According to theories, only odd harmonics can exist for the configuration of IDTs used here (D. Morgan, surface acoustic wave filters, Elsevier Ltd (2007)). This should be clarified.

Answer: We thank the reviewer's reminding and have carefully checked the resonance peak at 365.65 MHz, which is determined to be the longitudinal leaky SAW (LLSAW) confirmed by our FEM simulations (Fig. S3). As the reviewer pointed out, there were indeed only odd harmonics for our uniform IDT (D. Morgan, surface acoustic wave filters, Elsevier Ltd (2007)), such as the 1st (181.63 MHz) and 3rd (546.03 MHz) harmonic peaks. The IDTs were fabricated on piezoelectric 128° Y-cut 90°X-propagation LiNbO₃ substrate to

launch SAWs. The 1st and 3rd harmonic peaks are determined to be the Rayleigh SAW, and 2nd peak are identified as the 1st harmonic peak of LLSAW. LLSAW has a higher fundamental frequency (365.65 MHz) than the first-order Rayleigh SAW (181.63 MHz) when excited with the same IDT structure due to its higher phase velocity (Appl. Phys. Lett. 86, 024101 (2005)). The velocity of LLSAW can be typically twice as that of the Rayleigh SAW (Appl. Phys. Lett. 82, 3351 (2003)), bringing about a nearly twofold difference in fundamental frequency. We have mentioned this point and revised the corresponding statements in Page 5 Line 2 of the main text as “**Figure 1c shows the typical SAW reflection spectrum of the delay line, which contains three excited modes corresponding to the first Rayleigh SAW (181.63 MHz, R1), the longitudinal leaky SAW (LLSAW) (365.65 MHz, LL) and the third Rayleigh SAW (546.03 MHz, R3), confirmed by our finite element simulations (FEM) (Supplementary Fig. S3).**”. The added figure S3 is shown as follows:

Fig. S3 | Finite element simulations of the SAW modes based on 128° Y- 90° X LiNbO₃ substrate with $\lambda = 20 \mu\text{m}$. **a**, The simulated device admittance curve. There are three resonance peaks with frequencies around 179, 319 and 527 MHz, respectively. The third peak is relatively weak (see the inset). **b**, The deformation shape for the three SAW modes at resonant frequency. From left to right, they are the first Rayleigh SAW (R1), the longitudinal leaky SAW (LL)

and the third Rayleigh SAW (R3). **c,d**, The distribution of three normalized particle displacement components along the white dashed line for R1 (c) and LL (d). The insets show the local magnification of the deformation and white dotted lines extending from the electrode surface to the inside of the substrate. The distance from the electrode surface d is normalized through dividing by the wavelength λ . The propagation direction of SAW is defined as the $+x$ -axis. The displacements of R1 concentrate well at the surface (1 ~ 2 times of λ), while those of LL go deeply into the substrate.

Q4: “The skyrmions density increases gradually from 0 to $10 \times 10^5 \text{ mm}^{-2}$ with increasing P ...”, the density is high considering the skyrmion size in micrometer.

Answer: To make the skyrmion density more accuracy, we counted a much wider area of the film and modified figure 2b and corresponding expression in page #7 line #2 from the bottom: **The skyrmions density increases gradually from 0 to $8 \times 10^5 \text{ mm}^{-2}$ with increasing P from 19 to 23 dBm.**

Q5: In page 8 “evolves into the mixture of skyrmions and stipe domains...”, “stipe” should be “stripe”.

Answer: We have revised this mistake and checked the whole manuscript. We appreciate very much the insightful and careful review.

Q6: In this work, obvious thermal effects are found and utilized to assist the skyrmion creation. However, skyrmion creation through SAWs performed by T. Yokouchi et al. (Nat. Nanotechnol. 15, 361–366 (2020)) shows that very low temperature change is observed when RF voltages are inputted into IDTs. What’s possible reason for this difference?

Answer: The reviewer raised a very good point and remind us to check the remarkable temperature increasing in our SAW delay line. As we responded to Q3, we actually utilize LLSAW to assist the skyrmions creation. In contrast to the Rayleigh SAW, whose strain and energy concentrate well at the surface, LLSAW goes deeply into the solid and radiates energy into the substrate (Appl. Phys. Lett. 86, 024101 (2005), IEEE Ultrasonics Symposium Proceedings 1, 287-292 (1994), IEEE Ultrasonics Symposium Proceedings 1, 157-160 (1997)). Resultantly, there are large propagation losses in LLSAW, and the leaky power eventually dissipates in the form of heat, resulting in a remarkable temperature rise. The dissipated power P_{diss} can be quantitatively calculated by S parameters (IEEE Transactions on Ultrasonics, Ferroelectrics, and Frequency

Control 56, 2686-2692 (2009)):

$$P_{\text{diss}} = P_{\text{in}}(1 - |S_{21}|^2 - |S_{11}|^2)$$

where P_{in} is the input power by the RF signal generator. LLSAW and Rayleigh SAW have comparable S_{21} , but the S_{11} of LLSAW is significantly smaller, resulting in larger P_{diss} and more apparent thermal effect. Therefore, LLSAW can generate more remarkable thermal effect than Rayleigh SAW.

Rayleigh SAW has been applied by T. Yokouchi et al. (Nat. Nanotechnol. 15, 361–366 (2020)) to create skyrmions, which does not have obvious thermal effect as the reviewer mentioned. In our 1st and 3rd Rayleigh SAW, the results are also the same in Fig. 1d in the Main Text. As a result, this difference is attributed to different heat generation between LLSAW and Rayleigh SAW. We have mentioned this point and revised the corresponding statements in Page 5 Line 9 from the bottom of the main text as **“It is clear that the temperature in the magnetic films increases from 28.6 °C to about 38.6 °C as the increase of applied power from 17 dBm to 22 dBm at 365.65 MHz, where LLSAW goes deeply into the solid and radiates energy into the substrate³⁹. Resultantly, there are large propagation losses in LLSAW, and the leaky power eventually dissipates in the form of heat, resulting in a remarkable temperature rise. In contrast, the temperatures in the magnetic film change much smaller (below 1 °C) at the same range of power at 181.63 MHz and 546.03 MHz. Therefore, we note that the LLSAW at 365.65 MHz provide both the strain and thermal effect in our delay line, paving the way for the later creation of organized skyrmions.”**.

Q7: The pulse width of driving current and the velocity of skyrmion motion should be added in the manuscript.

Answer: The duration of current pulses is 50 μs and we have added it in the caption of figure 3. In addition, we have added the velocity of skyrmions motion in figure S10 as follows:

Fig. S10 | Current density dependence of average velocity of skyrmion motion generated by SAWs. When the current density is small ($J < 1.25 \times 10^{10} \text{ A m}^{-2}$), skyrmions cannot be driven by the current, which is due to the fact that a certain large of current is needed to overcome the skyrmions pinning barrier. As increasing current density from $1.25 \times 10^{10} \text{ A m}^{-2}$ to $8.3 \times 10^{10} \text{ A m}^{-2}$, skyrmions start to move and the corresponding velocity of skyrmion motion increases rapidly.

Response to Reviewer # 3

In this work, the authors demonstrate the ordered formation of skyrmions and the suppression of the skyrmion Hall angle via the application of surface acoustic waves (SAWs). The reduction of the skyrmion Hall angle is one of the main topics in the skyrmion research. Recently, there are several ways to reduce the skyrmion Hall angle have been proposed and experimentally demonstrated. Since utilizing SAWs to achieve this is a novel way, this work has a great impact on spintronics. However, as described below, there are several concerns and uncertainty in the present form of the manuscript, some of which may affect the main conclusion. Hence, if the authors can clarify the following concerns, I can recommend the publication.

Response: We are very grateful to the referee's support and the positive evaluation of our work. According to the referee's suggestions and requirements, we further revised our manuscript and clarified the proposed concerns in the revised version.

Q1: Did the authors use propagating SAWs or standing SAWs? Judging from Fig. 1(a) and (b), it seems propagating SAWs were used. If this is the case, what determines the positions of anti-nodes (i.e. the position of the skyrmion array?). In general, the

positions of the anti-nodes of propagating SAWs move with time.

Answer: The reviewer raised a very good point and remind us to clarify this point more clearly. According to our FEM simulations (Fig. S3 and S4), the longitudinal leaky SAW (LLSAW) has been excited when an RF voltage is applied to IDT 1 (Fig. S4d). Then, LLSAW propagates along the +x-axis and coherently superimposed with the reflected wave from the IDT 2 to form a bulk longitudinal wave (leakage along the -z-axis). Meanwhile, the strain distribution on the substrate surface has the characteristics of standing waves, i.e. there are some locations where the strain is always zero (nodes) or largest (anti-nodes), as shown in Fig. S4d. The distance of the two IDTs and the wavelength of LLSAW determine the positions of anti-nodes, which do not move with time. Along with these anti-nodes, skyrmions are confined near their positions and form an array eventually. We have revised Fig. 1a and mentioned this point in Supplementary Note 2 as **“The SAW admittance and transmission spectrums are simulated by commercial COMSOL 6.0 Multiphysics platform using piezoelectricity component based on solid mechanics and electrostatics. To reduce computational load, we build 2.5D resonator and acoustic delay line (ADL) model and set a proper thickness in plane. As for resonator simulation, the model possesses following geometry: wavelength $\lambda=20\ \mu\text{m}$, the lithium niobate (LN) substrate was set to be 4.5λ and 1λ perfect match layer was added to the bottom for capturing losses related to bulk wave radiation. In order to simplify the calculation time, only a period of IDT is considered, and periodic boundary condition has been set in both the x- and y-directions to simulate the infinite length. As for ADL simulations, the distance of the two IDTs is $200\ \mu\text{m}$, and each IDT has 20 pairs of single-type fingers. In each IDT, the width and gap of the fingers are both set to $5\ \mu\text{m}$, defining the fundamental wavelength of the SAW to be $20\ \mu\text{m}$. Perfectly matched layers (PML) are placed around the LN cells for the same effect as resonator. The frequency domain studies are carried out to calculate the Y- and S-parameters of resonator and ADL, respectively. The calculated S21 of this device showing three transmission maxima, in coincidence with those of the device used in the experiment. For the second peak which is used in our experiments, the LLSAW is excited when an RF voltage is applied to IDT 1 (Fig. S4a). Then, LLSAW propagates along the +x-axis and coherently superimposed with the reflected wave from the IDT 2 to form a bulk longitudinal wave (leakage along the -z-axis). Meanwhile, the strain distribution on the substrate surface has the characteristics of standing waves, i.e. there are some locations where**

the strain is always zero (nodes) and where the strain is largest (anti-nodes), as shown in Fig. S4d. Along with these anti-nodes, the position of the skyrmion array is determined.”. The added figure S3 and revised figure S4 are shown as follows:

Fig. S3 | Finite element simulations of the SAW modes based on 128° Y- 90° X LiNbO_3 substrate with $\lambda = 20 \mu\text{m}$. **a**, The simulated device admittance curve. There are three resonance peaks with frequencies around 179, 319 and 527 MHz, respectively. The third peak is relatively weak (see the inset). **b**, The deformation shape for the three SAW modes at resonant frequency. From left to right, they are the first Rayleigh SAW (R1), the longitudinal leaky SAW (LL) and the third Rayleigh SAW (R3). **c,d**, The distribution of three normalized particle displacement components along the white dashed line for R1 (c) and LL (d). The insets show the local magnification of the deformation and white dotted lines extending from the electrode surface to the inside of the substrate. The distance from the electrode surface d is normalized through dividing by the wavelength λ . The propagation direction of SAW is defined as the $+x$ -axis. The displacements of R1 concentrate well at the surface (1 ~ 2 times of λ), while those of LL go deeply into the substrate.

Fig. S4 | Finite element simulations of the SAW transmission spectrum in a delay line based on $128^\circ\text{Y-}90^\circ\text{X LiNbO}_3$ substrate. **a,b**, Device geometry and corresponding coordinate system. The distance of the two IDTs is $300\ \mu\text{m}$, and each IDT has 20 pairs of single-type fingers. In each IDT, the width and gap of the fingers are both set to $5\ \mu\text{m}$, defining the fundamental wavelength of the SAW to be $20\ \mu\text{m}$. **c**, The calculated S_{21} of this device, showing three transmission maxima, in coincidence with those of the device used in the experiment. **d**, The strain distribution near the surface of LiNbO_3 substrate in the xOz plane, showing the characteristics of standing waves.

Q2: Figure 3 shows the skyrmion Hall angle with and without applying SAWs. However, the initial conditions for these experiments differ. When both SAWs and the current are applied, skyrmions and stripe domains coexist in the initial state (Fig. 3a). In contrast, when only the current is applied, only skyrmions exist in the initial state (Fig. S5). It is important to note that stripe domains can affect the skyrmion motion and reduce the skyrmion Hall angle [see PRB 101, 214432 (2020)]. Hence, for a fairer comparison, the same initial state (hopefully initial state accommodating only skyrmions) should be used. For example, do the skyrmions created by the heater show the small skyrmion Hall angle if the current and SAW are applied simultaneously? If SAWs reduce the skyrmion angle as the authors concluded, the skyrmions created by the heater must show the small skyrmion Hall angle when both SAWs and current are applied.

Answer: We agree with the referee that the stripe domains contribute to the inhibition of transversal motion of skyrmions and this has been detailedly

discussed in previous work (Phys. Rev. B 101, 214432 (2020), Phys. Rev. Appl. 18, 024030 (2022)). In our paper, it is noted that both skyrmions and stripe domains are orderly aligned under the excitation of SAW as shown in figure 3a. This means not only skyrmions but also the stripe domains show more stable state at the anti-nodes of waves as discussed in the micromagnetic simulation part. Thus, it can be seen that the energy redistribution caused by SAW is the most fundamental reason that leads to the ordered alignment of skyrmions and stripe domains and eventually causes the negligible skyrmion Hall effect.

In addition, as the referee suggested, we also tried to perform the current driven measurement in the state with only skyrmions. However, it is very hard to obtain the state in which only skyrmions exist under the excitation of SAWs. Nevertheless, to verify the importance of SAWs to the ordered alignment feature, we showed the mixed states of skyrmions and stripe domains generated by the heater in figure S6 as comparison. One can obviously see that the skyrmions and stripe domains are randomly distributed without any ordered states, suggesting the significant role of SAW on the ordered alignments of skyrmions and stripe domains. In the main text, we added the discussions and corresponding references in page #10 line #6 from the bottom: **It is worth noting that not only skyrmions but also the stripe domains exhibit ordered alignment which can contribute to the inhibition of transversal movement of skyrmions^{41,42}. As comparison, we prepared a sample with the same structure in which the mixed states of skyrmions and stripe domains generated through a heater (Supplementary Fig. S6). One can obviously see that the skyrmions and stripe domains are randomly distributed without any ordered states, suggesting the significant role of SAW on the ordered alignments of skyrmions and stripe domains.** The added figure S6 is shown as follows:

Fig. S6 | MOKE image for the mixed states of skyrmions and stripe domains generated by thermal effect in the sample of Co/Pd/Co/Pd/Co/Pt. Scale bar, 10 μm .

Q3: In Fig. 3a, the authors show the snapshots of the magnetic texture when the current is applied. As can be seen from Fig. 3a, many bubbles (represented by red, orange, yellow, and blue circles) move along the current, supporting the authors' assertion that these bubbles are topologically nontrivial (i.e. skyrmion). However, a few bubbles (green and pink circles) are elongated (vi) and disappear (vii), which indicates these bubbles are topologically trivial (i.e. not skyrmion), because the topologically trivial bubbles are elongated or shrink under the current [Science 349, 283 (2015)]. Hence, a few trivial bubbles are probably created by SAWs. While this point does not affect the authors' main conclusion, the authors should discuss this point more carefully in the main text.

Answer: We think that the bubbles marked by different color circles are topologically nontrivial skyrmions. As shown in (i)-(iv) of figure 3a, the bubbles marked by green and pink circles can be driven and move along the current direction, which confirms their topologically nontrivial feature. The elongation and disappearance of these bubbles in (vi) and (vii) are most likely due to the pinning effect in the film which prevents the movement of skyrmions and eventually leads them to merge with other domains. In fact, there is an apparent defect of the film around the pink circle in (vi) and this may act as the pinning site. As suggested by the referee, this should be discussed in the main text and we have added the explanations in page #10 line #11: **We note that the elongation and disappearance of the bubbles marked by green and**

pink circles in (vi) and (vii) are most likely due to the pinning effect in the film which prevents the movement of skyrmions and eventually leads them to merge with other domains.

Q4: The authors have simulated the current driven motion of skyrmions with SAW, which seems qualitatively consistent with the experiment results (Fig. S7). However, I cannot find any explanation or citation of this result in the main text. If the result shown in Fig. S7 has not been described in the main text, it should be mentioned.

Answer: We appreciate the reviewer's reminding and have added the discussion of the results in figure S7 in page #11 line #1: **We also performed the micromagnetic simulations to study the current induced skyrmions motion with exciting SAWs. Consistent with the experimental results, skyrmions are first randomly distributed before exciting SAWs and then orderly aligned once the SAW is applied. Under the current pulses, skyrmions are moving in the current direction nearly in a straight line with negligible skyrmion Hall effect (Supplementary Fig. S7).**

Q5: How much is the gap size of the IDT used for exciting SAW with 2.79 GHz (Fig. S6)?

Answer: We thank the referee for this reminding. The width and gap of the fingers used to excite SAW with 2.79 GHz are 350 nm. We have added this information in the supplementary information.

Reviewers' Comments:

Reviewer #1:

Remarks to the Author:

The authors have addressed my comments in the first-round review. I have following further minor comments:

1. The ordered creation and motion of skyrmions are explained by the periodic strain distribution. It is well known that strain is a tensor consisting of six components. It is not yet clear which strain component plays a leading role in the formation of ordered skyrmions. In Figure S4 (d), it shows the strain distribution near the surface, but which strain component is not specified.

2. On page 3, the recent work on the manipulation of magnetic topological structure by SAWs should be included for integrity, for example, Vortex Core Reversal by Elastic Waves in Ferromagnetic Materials [International Journal of Solids and Structures 233: 111213 (2021)] and Motion of a magnetic skyrmionium driven by acoustic wave [Appl. Phys. Lett. 121, 242406 (2022)].

Reviewer #2:

Remarks to the Author:

The authors provide extra experiments, simulations, and discussions about the LLSAW and its role on the ordered skyrmion creation and motion. The experimental results and statements are clear. However, there are still some concerns need to be further clarified before the work can be considered for publication.

1) Whether one can expect the standing wave can also be formed in the Rayleigh waves? A supporting reference for the formation of LLSAW on 128° , rotated, Y-cut LiNbO₃ substrate would be helpful.

2) The current version is clear that the order domain structure is created by LLSAW, including the stripe domains. The inhabitation of the SkHE is then attributed to the SAW. While for the skyrmionics, the isolated skyrmion is preferred, so the authors should give some comments on that.

3) The abstract is suggested to be modified to fit the results and statements, especially for the inhabitation of the SkHE.

Reviewer #3:

Remarks to the Author:

In the revised manuscript, the authors have addressed all my concerns and suitably revised the manuscript. Hence, I recommend the publication of the manuscript in Nature Communications.

Response Letter of NCOMMS-22-52692A

Response to Reviewer # 1

The authors have addressed my comments in the first-round review. I have following further minor comments:

Response: We are very grateful to the reviewer's support. According to the review's suggestions and requirements, we further revised our manuscript and gave more in-depth discussions in the revised version.

Q1: The ordered creation and motion of skyrmions are explained by the periodic strain distribution. It is well known that strain is a tensor consisting of six components. It is not yet clear which strain component plays a leading role in the formation of ordered skyrmions. In Figure S4 (d), it shows the strain distribution near the surface, but which strain component is not specified.

Answer: The reviewer raised a very good point. The longitudinal strain ϵ_{xx} in LLSAW plays a leading role in the formation of ordered skyrmions. LLSAW possesses displacement components in the x and z directions, while the transverse component u_y is zero (Appl. Phys. Lett. **82**, 3351 (2003), Appl. Phys. Lett. **86**, 024101 (2005)). Only ϵ_{xx} , ϵ_{zz} and ϵ_{xz} exist among the six independent components of the strain tensor. The longitudinal displacement u_x is considered to be the dominating component in LLSAW (Appl. Phys. Lett. **86**, 024101 (2005)), which indicates that ϵ_{xx} is the dominant one among the three nonvanishing strain components and this is also confirmed by our FEM simulations. Therefore, the magnetoelastic coupling energy is mainly contributed by ϵ_{xx} , which plays a leading role in the ordered alignment of skyrmions. In Fig. S4d, we show the spatial distribution of ϵ_{xx} near the surface of LiNbO₃ substrate in the xOz plane. We have included this point in the caption of Fig. S4d.

Q2: On page 3, the recent work on the manipulation of magnetic topological structure by SAWs should be included for integrity, for example, Vortex Core Reversal by Elastic Waves in Ferromagnetic Materials [International Journal of Solids and Structures 233: 111213 (2021)] and Motion of a magnetic skyrmionium driven by acoustic wave [Appl. Phys. Lett. 121, 242406 (2022)].

Answer: We thank the reviewer for this valuable suggestion and recommending these important references to us. We have included them and other relevant literature in our revised manuscript as references 38–42.

Response to Reviewer # 2

The authors provide extra experiments, simulations, and discussions about the LLSAW and its role on the ordered skyrmion creation and motion. The experimental results and statements are clear. However, there are still some concerns need to be further clarified before the work can be considered for publication.

Response: We are grateful to the reviewer's support and the positive evaluation of our work.

Q1: Whether one can expect the standing wave can also be formed in the Rayleigh waves? A supporting reference for the formation of LLSAW on 128° , rotated, Y-cut LiNbO₃ substrate would be helpful.

Answer: The reviewer raised a very good point and we thank the reviewer for this reminding. According to our FEM simulations (Fig. R1), the standing wave can also be formed in the Rayleigh wave by the same coherent superposition process as LLSAW. The positions of nodes (the white lines) do not advance forward as time increases, exhibiting the feature of the standing wave. We have added references 44–46 about the formation of LLSAW on 128° Y-cut LiNbO₃.

Fig. R1 | The evolution of strain ϵ_{xx} from t to $t+T/5$ in the first-order Rayleigh wave (R1). T is the vibration period of this SAW. The positions of anti-nodes (red and blue) and nodes (white) do not move with time, showing the characteristics of the standing wave.

Q2: The current version is clear that the order domain structure is created by LLSAW, including the stripe domains. The inhibition of the SkHE is then attributed to the SAW. While for the skyrmionics, the isolated skyrmion is preferred, so the authors should give some comments on that.

Answer: Many thanks to the reviewer's comments. We agree with the reviewer that the isolated skyrmion is preferred for the future skyrmionics. In this work, we focus on the inhibition of SkHE. By applying LLSAW to ferromagnetic multilayers, we have successfully realized the ordered creation and motion of skyrmions with negligible SkHE. As the reviewer pointed out, the isolated skyrmion and its movement with negligible SkHE are the ultimate goal to pursue for practical application. In further research, this goal is expected to be achieved through the modulation of SAW power as well as various modes, combined with the optimization of material system, the design of device configuration and so on, eventually advancing the progress of further skyrmion-based spintronic devices.

Q3: The abstract is suggested to be modified to fit the results and statements, especially for the inhibition of the SkHE.

Answer: We thank the reviewer's valuable suggestion. We have mentioned this point and revised the corresponding statements in the abstract as **"Here, by embedding the [Co/Pd] multilayer into a surface acoustic wave (SAW) delay line where the longitudinal leaky SAW is excited to provide both the strain and thermal effect, we experimentally realized the ordered generation of magnetic skyrmions. The resultant current-induced skyrmions movement with negligible SkHE was observed, which can be attributed to the energy redistribution of the system during the excitation of SAW."**

Response to Reviewer # 3

In the revised manuscript, the authors have addressed all my concerns and suitably revised the manuscript. Hence, I recommend the publication of the manuscript in Nature Communications.

Response: We are very grateful to the reviewer's support and the positive evaluation of our work.

Reviewers' Comments:

Reviewer #1:

Remarks to the Author:

The revised version is publishable.

Reviewer #2:

Remarks to the Author:

All my comments and concerns have been carefully addressed and revised in the manuscript by authors. So I recommend the publication of the current version in Nature Communications.